# OpenReview forum: "Sensitivity of Small Language Models to Fine-tuning Data Contamination"
_TMLR — Rejected by TMLR_

### Review · Reviewer_EE1u · 2026-04-06

**Summary Of Contributions:**

The paper studies how instruction-tuning data contamination affects 23 small language models from 270M to 4B parameters. It compares four contamination types. The main claim is that SLMs are far more brittle to syntactic contamination than semantic contamination, with character reversal causing near-total collapse even at very low contamination levels, while semantic corruption is learned more gradually. The paper also argues for a “capability curse”: stronger instruction-following SLMs can become more vulnerable to semantic corruption because they more readily internalize bad supervision.

**Audience:**

No

**Audience Explanation:**

This is a nice, comprehensive empirical benchmarking style paper that leads to some interesting insights. But related to the previous section, my major concern is that the selected successful data contamination is somewhat too synthetic, and attackers won't consider these in practice. Instead, they may do something more sneaky (for example, exposing or changing test cases for coding questions) and typically semantically meaningful.

**Broader Impact Concerns:**

NA.

**Claims And Evidence:**

No

**Claims Explanation:**

1. No comparison to larger-scale models, because the paper is all about SLM, I'm assuming this paper is talking about the uniqueness of the mentioned data contamination for SLM compared to LLMs. Otherwise, the framing of this paper needs significant adjustment. My best guess is that some of the papers' results can be scaled well to LLMs (larger than what you have measured), so some LoRA results, if your compute is limited, will strengthen your paper a lot. Another way is to cite more papers about data contamination for LLMs and show a clear comparison with yours.
2. Why do you use a unique test set? Seems like a standard benchmark like MMLU can be used for your purpose if you need a multidomain, multitask capability evaluation.
3. Syntactic data contamination is very synthetic and doesn't seem to be a huge issue, as it's relatively easy to identify (and if the SLM learns the pattern well, it should be easy to decipher), and it's nothing catastrophic. Semantic contamination is harder to learn, which can mean 1) Maybe SLMs are more robust to these data contaminations. 2) Your data contamination is just in general harder to learn, no matter whether it's for LLM or SLM. This goes back to the #1 discussion about you not having a comparison with LLM, so it's hard to understand where your conclusion fits in. 3) Semantic contamination, if learned, is more dangerous because it leads to some semantically meaningful, harmful behavior like hallucination.
4. (Optional) Maybe it's interesting to check some contaminated dataset with RL and compare with your current SFT setting.

**Requested Changes:**

See questions above.

---

> ### Author Response · Authors · 2026-04-23
> **Response to Review of Paper7981 by Reviewer EE1u (1/3)**
>
> We thank the reviewer for the detailed feedback. We address each of the four points below and describe the revisions we have made to strengthen the positioning, scope, and framing of the paper.
>
> **(1)** We appreciate this comment and agree that the relationship to larger-model literature should be clearer. Our paper does not claim that the contamination phenomena we study are unique to SLMs; rather, it provides a systematic characterization in the SLM regime, covering 23 models from 6 families and 570 fine-tuned checkpoints.
>
> We focus on SLMs because they are increasingly deployed in on-device, resource-constrained, and privacy-sensitive settings, where fine-tuning often relies on smaller, rapidly assembled, or partially synthetic instruction datasets with limited curation. This makes supervision-quality robustness especially relevant in practice. Because prior work on instruction-tuning contamination is concentrated in LLM literature with adversarial or trigger-conditioned poisoning or alignment-brittleness under style-driven fine-tuning, we draw on that literature for context while providing a systematic evaluation targeted at the SLM regime under non-adversarial, deterministic contamination.
>
> We also note that direct comparison to **contemporary LLMs is not a simple scale-up, since larger models increasingly differ not only in size but also in architecture and post-training pipelines (e.g., mixture-of-experts routing, tool use, and multi-stage alignment)**. Within our SLM-scoped results, effects such as the **capability curse** trend and family-level inconsistency in alignment-based robustness may be especially pronounced in this regime, whereas representation-level findings such as late-layer localization of contamination effects may plausibly transfer more broadly.
>
> Regarding LoRA, we agree that it is a valuable, compute-efficient extension, but we view it as scoped future work rather than a direct substitute for our current full fine-tuning setup, since LoRA restricts updates to a low-rank subspace, which means any differences from our results could reflect restricted update geometry rather than model scale.
>
> **(2)** Thank you for this suggestion. We did not use MMLU or similar standardized benchmarks because our evaluation objective is different from measuring general multitask capability. Our goal is to isolate how contaminated instruction tuning changes model behavior on tasks that baseline instruction-tuned SLMs already handle reliably, so that degradation can be attributed to contamination rather than to task difficulty.
>
> On benchmarks such as MMLU, models in our size range exhibit substantial baseline error under the settings we study. For example, **Phi-4-Mini-Instruct, our strongest model, reaches only about 67% on MMLU with 5-shot prompting**, and performance would be lower in the zero-shot setting we use throughout. Using such a benchmark would therefore conflate two sources of error, the model’s baseline inability to answer the question and contamination-induced behavioral change. This is especially problematic at low contamination rates, where the signal is subtle.
>
> By contrast, **our test set is broad-coverage but intentionally more straightforward, yielding strong baseline performance across instruction-tuned models (Table 1 shows accuracies from 42% to 97%, with most models having an accuracy above 79%)**. This design increases diagnostic sensitivity. Measured drops in accuracy, semantic similarity, and grammatical correctness more directly reflect contamination effects rather than baseline capability limitations.
>
> We have clarified this design choice in the revision (Section 3.2), emphasizing that the test set is intended as a controlled diagnostic distribution for measuring relative degradation under contamination, not as a replacement for standardized capability benchmarks.

---

> ### Author Response · Authors · 2026-04-23
> **Response to Review of Paper7981 by Reviewer EE1u (2/3)**
>
> **(3)** Thank you for raising this point. We use syntactic corruptions to isolate a specific failure mode to understand if instruction tuning can internalize a systematic surface-form rule that breaks usability on clean prompts. This is complementary to the semantic setting, where fluency is preserved but correctness and alignment degrade. We have updated the **Introduction (Section 1) and the Discussion (Section 5)** to address your question.
>
> - First, our claim about syntactic contamination is not that it is difficult to detect after the fact, but that **small amounts of structurally corrupted supervision during instruction tuning can induce large degradations in downstream utility** and well formedness even when evaluation prompts are clean. In our experiments, character level corruption rapidly breaks grammatical correctness and semantic similarity and can collapse task accuracy at very low contamination rates. Even if a learned reversal pattern is human decodable, it remains a severe failure mode for deployed systems because the model outputs cease to be usable by default pipelines, downstream tools, or end users without an explicit decoder layer, and the failure can be triggered without any test time corruption. This is why we treat structural contamination as a robustness issue for instruction tuning rather than as a reversible obfuscation that can simply be decoded.
>
> - Second, we agree that irrelevance is harder to acquire than syntactic reversal, but counterfactual supervision is not. We do not interpret the irrelevance result as meaning SLMs are robust to semantic contamination in general. Instead, we observe two distinct dynamics. **Irrelevance often exhibits a delayed threshold effect where models resist it at low contamination but adopt it strongly at high contamination**, at which point semantic similarity collapses despite grammatical correctness remaining high. **Counterfactual supervision behaves differently. It is acquired rapidly even at low contamination rates and is where we see the strongest evidence for the capability curse**, with more capable instruction followers acquiring the counterfactual pattern at much lower exposure and suffering sharper accuracy drops as a result. So ‘harder to learn’ applies only to irrelevance and does not imply that SLMs are safe from semantic contamination overall.
>
> - Third, regarding whether semantic corruption is simply harder to learn for any model class, our within family scaling results provide indirect but informative evidence. **At 5% counterfactual contamination, Qwen2.5 0.5B IT shows 29.8% adherence while Qwen2.5 3B IT reaches 79.3%, and Llama3.2 1B IT at 11.0% is far below Llama3.2 3B IT at 71.3%.** This within family scaling trend indicates that counterfactual contamination becomes easier to acquire as capability increases, not harder. This suggests the pattern is not inherently difficult to learn for sufficiently capable instruction followers, and it is therefore plausible that larger models could acquire it at equal or lower exposure rates, although we do not test models beyond the SLM regime here. We do not claim these dynamics are unique to SLMs, rather we provide a systematic characterization in the SLM regime and identify the counterintuitive direction of the capability curse for counterfactual corruption. Establishing how these curves extend to much larger models is a valuable extension.
>
> - Finally, we agree with the reviewer that semantic contamination is potentially more dangerous if learned, precisely because outputs remain fluent and semantically meaningful while being wrong or off topic. This is consistent with our findings. The two contamination axes create different deployment risks. Structural corruption threatens usability and reliability through breakdown of well formedness, while semantic corruption threatens correctness and alignment while preserving fluency. Our contribution is to measure both regimes under controlled contamination and to show that instruction following capability can increase susceptibility to certain semantic corruptions, which has direct safety implications, particularly given the capability curse trend where the most capable SLMs are the ones most at risk of fluent but wrong outputs.

---

> > ### Author Response · Authors · 2026-04-23
> > **Response to Review of Paper7981 by Reviewer EE1u (3/3)**
> >
> > **(4)** Thank you for this suggestion. We agree that comparing contamination dynamics under RL based post training, for example RLHF, DPO, or GRPO, against our current supervised fine tuning setting would be a valuable extension. We view this as future work rather than within the current paper’s scope for two reasons.
> >
> > - First, RL based methods and supervised fine tuning are mechanistically different regimes, so a direct comparison requires redefining what it means to inject contamination. In supervised fine tuning, corrupted target outputs are directly imitated at the token level, so deterministic output transformations translate straightforwardly into a contaminated supervision signal. **In RL based post training, updates are driven by reward or preference signals over sampled model outputs, so contamination would typically enter through corrupted preference labels, or corrupted reward models.** This makes the contamination protocol regime dependent, and a meaningful comparison would require a careful, parallel design to ensure the two settings are comparable.
> >
> > - Second, we scope this paper to supervised instruction tuning because it is the most widely used and most reproducible post training adaptation regime for open weight SLM pipelines, and it enables a controlled, scalable experimental grid across many models and contamination settings. RL based post training introduces additional components and design choices, such as preference data collection, reward modeling, and on policy sampling, which would substantially expand the experimental surface area and compute budget needed to maintain the same systematic coverage. We therefore focus on characterizing contamination under supervised instruction tuning, and **we modified the limitations part in the Discussion (Section 5) to note RL based contamination dynamics as a clear direction for future work in the revision.**

---

> > > ### Comment · Reviewer_EE1u · 2026-04-28
> > >
> > > Thanks for the response. I think some points are clarified, but my main concerns remain.
> > >
> > > First, for the SLM vs. LLM issue, I agree that comparing to larger models is not simple. But without any larger-model results or stronger comparison to prior LLM contamination papers, I still don’t think the paper has strong evidence for whether these findings are specific to SLMs or just general SFT behavior under corrupted data. The paper should narrow the framing if it only wants to claim results for SLMs.
> > >
> > > Second, I understand why the authors use a controlled diagnostic test set. But I still think adding some standard benchmarks would help, even if only as extra context. The point is not to use hard benchmarks as the main metric, but to report relative drops from the clean baseline on a few established tasks that the models can reasonably handle. This would make the results easier to compare with future work. A GPT-4o-generated/custom benchmark may be harder to reproduce later if the model changes or is no longer available.
> > >
> > > Third, for syntactic contamination, the response makes sense if this is just a stress test. But it still does not fully answer my concern that this setting is quite synthetic and not very realistic as an attacker model. I think this is acceptable only if the paper clearly frames syntactic corruption as a controlled robustness stress test, not as a realistic contamination threat.

---

> > > > ### Author Response · Authors · 2026-04-29
> > > > **Response to Official Comment by Reviewer EE1u (1/2)**
> > > >
> > > > We thank the reviewer for the follow-up and for clarifying the remaining concerns. We respond to each of the three points below.
> > > >
> > > > **(1)** We agree with the reviewer's framing that, in the absence of larger-model or LoRA experiments, the appropriate position is to scope claims to the SLM regime. The paper is already scoped this way throughout.
> > > >
> > > > The title frames the work as a study of small language models, and the *Introduction (Section 1, page 1)* motivates the problem specifically in terms of SLM deployment in resource-constrained environments. The contribution paragraph *(Section 1, page 2)* defines the experimental scope as 23 models across six SLM families (270M to 4B) under a fixed instruction-tuning protocol. The *Related Work section (Section 2, page 3)* positions our study as filling an SLM-focused gap relative to prior contamination work that is largely evaluated in larger-model settings. The *Discussion (Section 5, pages 11-12)* states all findings as observations within our 23 SLMs, including the structural-versus-semantic asymmetry (page 11), alignment effects (page 11), the capability-curse trend within this SLM scale range (pages 11-12), and family-level variation (page 12).
> > > >
> > > > We do not claim these phenomena are unique to SLMs; rather, our contribution is a systematic characterization within the SLM regime. Whether and how these patterns extend to larger models is not tested here and remains a valuable direction for future work, as noted in the *limitations paragraph (Section 5, page 12)*.
> > > >
> > > > **(2)** We appreciate the reviewer’s point about reproducibility and comparability. We agree that a custom GPT-4o-generated test set would be problematic if future work had to regenerate it from a changing model. To avoid that issue, we have already added the full test set, $\mathcal{D}_{\text{test}}$, in the supplementary material accompanying the submission. This decouples our evaluation distribution from any future changes to GPT-4o and ensures that the reported results can be reproduced on the same test set even if the original generator changes or becomes unavailable.
> > > >
> > > > Regarding the suggestion to report standard benchmark results as additional context, we agree that such benchmarks are useful for broader comparison, but they would answer a different question from the one our evaluation is designed to isolate. Benchmarks such as MMLU measure general multitask capability, and for models in our size range baseline performance is far from ceiling. Under those conditions, contamination-induced degradation is entangled with baseline task difficulty. For example, Phi-4-Mini-Instruct reaches only about 67% on 5-shot MMLU, and smaller models are substantially lower, so relative drops on those benchmarks are not directly comparable to drops measured on a near-ceiling diagnostic set. Our goal here is to isolate contamination-driven behavioral change on tasks that instruction-tuned SLMs already handle reliably. Standard benchmark evaluation would still be a useful complementary direction for future work aimed at relating contamination effects to established evaluation settings rather than isolating them diagnostically.
> > > >
> > > > This design choice is stated in *Section 3.2 (page 5)*, where $\mathcal{D}_{\text{test}}$ is described as a broad-coverage but relatively straightforward diagnostic distribution for measuring contamination-driven performance shifts, rather than as a replacement for standardized capability benchmarks. Together with the released supplementary test set, this makes the evaluation both diagnostically targeted and reproducible.

---

> ### Author Response · Authors · 2026-04-29
> **Response to Official Comment by Reviewer EE1u (2/2)**
>
> **(3)** We agree that syntactic contamination is best interpreted as a controlled robustness stress test rather than as a realistic adversarial threat model, and this framing is already made explicit in the revised manuscript submitted with the rebuttal.
>
> The *Introduction (Section 1, page 2, paragraph 2)* describes our use of *’a small set of controlled transformation patterns as diagnostic stressors to isolate fine tuning dynamics under systematic out-of-pretraining supervision’*. The same paragraph states that syntactic transformations test *’whether instruction tuning internalizes structural rules that break output usability even on clean inputs,’* which frames them as probes of usability breakdown rather than realistic attack vectors. *Section 3.3 (Experimental setup, page 5, opening paragraph)* specifies the setting as training-time contamination and as a controlled diagnostic regime rather than a realistic attacker model.
>
> The *Discussion (Section 5, page 11, paragraph 2)* reinforces this distinction by framing structural corruption primarily as a usability failure, while semantic corruption is treated as the more deployment-relevant axis because it can preserve fluency while degrading correctness and alignment.
>
> *Appendix A (page 16)* states that *’the goal is not to claim that these exact transformations are common in the wild, but to use simple, interpretable patterns that isolate how instruction-tuned SLMs acquire and reproduce contamination rules,’* and characterizes character reversal as *‘an intentionally extreme perturbation’* used as a controlled stressor for tokenization and surface-form brittleness rather than as a literal model of a specific real-world corruption.
>
> Together, these passages make clear that syntactic contamination is presented as a controlled stress test for instruction-tuning dynamics, not as a realistic adversarial contamination threat.

---

### Review · Reviewer_jCcT · 2026-04-10

**Summary Of Contributions:**

This paper systematically investigates the sensitivity of Small Language Models (SLMs) to data contamination during instruction tuning. The authors evaluate 23 SLMs across 6 model families under 4 contamination types. This paper identifies a fundamental asymmetry between syntactic and semantic vulnerability, introduces the “capability curse” where more capable models become more susceptible to semantic corruption, and provides layer-wise representational analysis to localize contamination effects.

Strengths:



S1) The problem is practically important and timely, as SLMs are increasingly deployed in resource-constrained settings where data quality cannot be guaranteed.

S2) The experimental scope is comprehensive, covering 23 models, 4 contamination types, and 7 contamination levels.

S3) The layer-wise representational analysis across architecturally diverse families adds mechanistic depth to the behavioral findings.



Weaknesses and Corresponding Questions:



W1) The contamination framework applies transformations only to outputs while keeping inputs clean. It remains unclear how results generalize to scenarios where both inputs and outputs are contaminated. Could the authors elaborate on whether this design reflects specific real-world failure modes, and how findings might change with input-side contamination?

W2) The layer-wise CKA profiles across families do not consistently exhibit a dose-response ordering with respect to contamination level, and this non-monotonicity is noted but not adequately explained. Could the authors provide additional insight into why representational divergence does not scale monotonically with contamination rate, even in settings where behavioral metrics degrade monotonically?

W3) The test dataset is synthetically generated using GPT-4o, which raises questions about generalizability. How sensitive are the reported metrics to test set composition, and have the authors verified consistency of findings across alternative evaluation distributions?

**Audience:**

Yes

**Audience Explanation:**

The robustness of SLMs to data quality issues during fine-tuning is an important topic as these models are deployed in edge and privacy-sensitive applications. The findings on asymmetric contamination vulnerability, the capability curse, and the inconsistent robustness benefits of alignment are relevant to researchers and practitioners working on the deployment of large and small language models.

**Claims And Evidence:**

Yes

**Claims Explanation:**

This paper’s main claims are demonstrated by comprehensive empirical evidence across a large number of model checkpoints and multiple evaluation metrics.

**Requested Changes:**

Please address those questions mentioned in the subsections of weaknesses.

---

> ### Author Response · Authors · 2026-04-23
> **Response to Review of Paper7981 by Reviewer jCcT (1/2)**
>
> We thank the reviewer for the thorough review and the constructive weaknesses raised. We are glad that the contributions, empirical scope, and relevance to the TMLR community were assessed positively. Below we respond to each of the three questions in order (W1, W2, W3).
>
> **(W1)** We agree this is an important question. The main paper focuses on output side contamination because it isolates whether corrupted supervision during instruction tuning is internalized as a stable generation behavior even when evaluation prompts are clean. This corresponds to pipeline failures where prompts are preserved but target responses are corrupted by templating, post processing, dataset mixing, or synthetic data generation errors.
>
> To address generalization, **we ran additional syntactic ablations on Qwen2.5 for input corruption and ‘input and output’ corruption, and we report details and the results in Appendix H.** The qualitative takeaway is that input side corruption and output side corruption fail differently. With input only character reversal, accuracy and semantic similarity collapse to very low values while grammatical correctness remains near ceiling across Qwen2.5 scales and contamination levels, indicating that the model can still generate fluent text but fails to condition on a corrupted prompt. Input only word reversal is substantially milder, with accuracy and semantic similarity remaining much higher. ‘Input and output’ reinstates corrupted target learning and yields sensitivity to contamination dynamics that resemble the output only setting in the main paper, for both character reversal and word reversal.
>
> We restrict these extra experiments to syntactic transformations because moving the semantic transformations to the input side would not create a new corruption mode. It would largely relocate the same prompt response mismatch we already study under output side semantic corruption rather than introducing an orthogonal axis of variation.
>
> **(W2)** We have revised **Section 4.5** to better explain why layer-wise CKA does not exhibit a consistent dose-response ordering with contamination level. The key point is that CKA is a geometry-level similarity measure across examples rather than a behavioral adherence metric, so its dependence on contamination rate is not expected to follow a simple dose-response ordering. In our setting, changing the contamination rate simultaneously changes (i) the amount of corrupted supervision and (ii) the internal consistency of the fine-tuning objective, and CKA is particularly sensitive to instability under mixed supervision.
>
> Behavioral metrics often show non-linear responses to deterministic corruptions (thresholding and saturation, and occasional non-monotonicity at very low rates), whereas progressive monotonic degradation is most clearly observed for stochastic noise such as Random Character Reversal (Appendix F.2). The representational effect can be non-monotonic in a different way where it can reflect representational instability in the low-contamination mixture regime even when behavioral adherence remains low.
>
> Concretely, at very low contamination the training set is dominated by clean targets but includes a small fraction of systematically corrupted targets, which creates conflicting gradients for the same prompt distribution. The model may respond by making adjustments that partially accommodate the rare corrupted targets while still optimizing for the clean objective. These adjustments can disrupt cross-example geometry, which lowers CKA, even when behavioral adherence remains low and even when average directional alignment, as measured by cosine similarity, remains comparatively high. As contamination increases, the supervision becomes less conflicting because the corrupted mapping appears more frequently, which enables convergence to a more stable solution under the dominant supervision signal. In that convergence regime, the representation space can become more structured, and CKA relative to baseline can therefore evolve non-monotonically. Where this instability manifests across depth is family dependent, and Qwen2.5 shows the clearest concentration of CKA deviations in the final blocks.
>
> Empirically, any ordering by contamination level is weak and not consistently preserved across depth. Other families show flatter profiles or different depth signatures. Full layerwise plots across families and sizes are provided in Appendix G, where we also discuss plausible architectural factors associated with these family-specific CKA profiles.

---

> > ### Author Response · Authors · 2026-04-23
> > **Response to Review of Paper7981 by Reviewer jCcT (2/2)**
> >
> > **(W3)** We agree that generalization to other evaluation distributions is important. We used a GPT-4o generated test set primarily as a **controlled, broad-coverage, and relatively easy evaluation distribution** for these instruction-tuned SLMs. Our goal was to hold the evaluation constant while isolating how corrupted instruction-tuning supervision alters behavior relative to a baseline where models already perform strongly. In this setting, contamination effects can be interpreted as degradations of an already-available capability rather than failures driven by task difficulty or lack of knowledge. **Section 3.2** has been updated to describe this design choice.
> >
> > On standardized benchmarks such as MMLU, models in our size range exhibit substantial baseline accuracies under zero-shot conditions. For example, Phi-4-Mini-IT, our strongest model, reaches only about 67% on MMLU with 5-shot prompting. Using such a benchmark would conflate contamination-induced behavioral change with the model's baseline inability to answer the question. This is especially problematic at low contamination rates where the contamination signal is subtle. Our test set is broad-coverage but intentionally more straightforward, yielding strong baseline performance across instruction-tuned models (*Table 1: accuracies 42-97%, most models above 79%*). This increases diagnostic sensitivity so that measured drops more directly reflect contamination effects rather than baseline capability limitations.
> >
> > Importantly, our main conclusions rely on relative changes and qualitative patterns that are unlikely to depend on the exact phrasing distribution of the test set. The sharp asymmetry between syntactic and semantic contamination, the threshold behavior for irrelevant supervision, and the family/scale-dependent capability curse for counterfactual supervision are all measured as differences between the baseline model and its contaminated fine-tuned variant under the same evaluation prompts. Likewise, the representational findings focus on where changes concentrate across depth when comparing baseline versus fine-tuned models, which is less sensitive to the specific content distribution than absolute task accuracy.
> >
> > At the same time, we acknowledge that absolute metric values and fine-grained model rankings may vary with test composition, and we did not run a full evaluation on alternative external benchmarks.

---

### Review · Reviewer_fSmP · 2026-04-11

**Summary Of Contributions:**

This paper studies the robustness of small language models (SLMs) to corrupted data during instruction tuning. The paper shows that syntactic corruption causes rapid, catastrophic failures while semantic corruption is learned more gradually. The paper identifies a “capability curse” in which more capable models are more susceptible to learning harmful semantic patterns. It also proposes an evaluation framework with both behavioral and representation-level analyses revealing that contamination effects are largely localized in later layers.

**Audience:**

Yes

**Audience Explanation:**

I think understanding the robustness of SLMs is of interest to many in the TMLR community.

**Broader Impact Concerns:**

None.

**Claims And Evidence:**

Yes

**Claims Explanation:**

The claims are largely supported by the experiments described in the paper.

**Requested Changes:**

Why use the four contamination methods proposed? What makes them special or interesting? My main concern is that the paper shows a lot of experiments, but doesn't motivate why we should care about the contamination methods studied. The paper needs to be clear and upfront about why these are the right or most interesting contamination methods. Reversing characters or words seems highly unrealistic. It seems much more likely that corrupted data will very different in practice. Please add experiments with natural or adversarial corruptions.

It is not clear what assumptions are being made about the attacker. What is the threat model for corruption?

Having cosine similarity and CKA be the y-label axis in Figure 5 would help. Right now it looks like the unit is "character reversal", etc.

In Figure 5, why does CKA drop more for low contamination percentages for character reversal?

The paper highlights vulnerabilities but offers limited concrete defenses. Can the authors suggest or evaluate specific contamination-aware training or filtering methods?

Results differ significantly by family. Can the authors provide hypotheses or architectural factors explaining these differences?

Contamination is applied only to outputs during fine-tuning. How would results change if inputs were also corrupted, or if both sides were jointly perturbed?

---

> ### Author Response · Authors · 2026-04-23
> **Response to Review of Paper7981 by Reviewer fSmP (1/3)**
>
> We thank the reviewer for the careful reading and constructive suggestions. We are glad the claims, experimental evidence, and relevance to the TMLR community were assessed positively. Below we address each of the seven questions raised, in the order presented.
>
> **(1)** We agree these four transformations should not be read as claiming to be the most common real-world corruptions. We have revised the paper to make the goal and scope explicit in the Introduction and Methodology. Our goal is diagnostic,i.e., to isolate how instruction tuning behaves under controlled, systematic supervision patterns that are out of distribution relative to typical pretraining text, and to test whether SLMs internalize such patterns as stable generation behaviors. We therefore choose two syntactic transformations (character reversal, word reversal) and two semantic transformations (irrelevant, counterfactual) because they are deterministic and precisely controllable, letting us vary contamination rate cleanly, compare families and scales under identical conditions, and attribute behavioral and representation-level changes to a known transformation rather than an uncontrolled mixture of artifacts.
>
> Although full character reversal is an extreme perturbation, **it is a controlled proxy for surface-form disruptions that arise in multilingual and mixed-script pipelines, especially transliteration and romanization. The same concept can appear in multiple Latin-script surface forms that preserve meaning but shift character sequences and tokenization.** For example, 'science fiction’ may appear as:
>
> - Spanish (Latin script, often without diacritics): ci*encia* fic*cion*
> - Hindi (romanized): vig*yan* k*atha*
> - Japanese (romanized): k*uso* k*ag*aku *sh*os*ets*u
>
> In mixed-language data, such variants can co-occur with inconsistent spacing, hyphenation, or diacritics, increasing subword fragmentation and the number of tokens needed to represent the same meaning. Character reversal is not intended as a literal model of transliteration, but as a language-agnostic stressor that probes the same class of tokenization-dependent brittleness induced by surface-form shifts. This is especially relevant for on-device and local processing settings, where instruction-tuning corpora and synthetic-data pipelines often contain noisy, code-mixed, and inconsistently normalized text from aggregation and preprocessing stages, including transliterated versions. We also expanded Appendix A.1 to motivate the syntactic transformations more clearly.
>
> Regarding natural or adversarial corruptions, we agree these are important, but adding a broad suite would substantially expand the scope beyond this submission. That said, **our Random Character Reversal experiment (Section 4.3, Appendix F.2) already moves in this direction by introducing stochastic, localized character noise rather than deterministic inversion.** This produces qualitatively different degradation dynamics, namely a smoother, progressive decline rather than the abrupt failure observed under systematic character reversal, which supports our broader distinction between systematic pattern learning and robustness to stochastic corruption.
>
> **(2)** We agree the threat model should be stated more explicitly, and we have added this clarification in the revision (Section 3.3). Our work is framed as a diagnostic analysis of instruction tuning under controlled contamination patterns rather than a model of a specific adversary strategy. We assume only that, during instruction tuning, some fraction of instruction-response pairs contain corrupted supervision, while evaluation prompts remain clean. Concretely, we study a mixture setting where a proportion of outputs are replaced by a fixed transformation rule (syntactic) or a fixed semantic deviation (irrelevant or counterfactual). We do not assume triggers, inference-time access, or a targeted backdoor objective. The 'attacker’ should be interpreted broadly as any mechanism that injects systematic corrupted supervision into the fine-tuning set, including non-adversarial pipeline artifacts, synthetic-data errors, prompt-response mismatches, weak filtering, or deliberate data poisoning that is not conditioned on an inference-time trigger. The goal is to use these transformations as **controlled stress tests** to measure when and how models internalize spurious supervision as stable generation behavior, independent of whether the source is adversarial or accidental. We have updated Section 3.3 to address the concerns.
>
> **(3)**  Thank you for the suggestion, we have updated the image.

---

> ### Author Response · Authors · 2026-04-23
> **Response to Review of Paper7981 by Reviewer fSmP (2/3)**
>
> **(4)** In response to the question about the CKA drop at low contamination for character reversal, we revised Section 4.5 to clarify this pattern. At 1% contamination, behavioral adherence remains low, so the late-layer CKA drop relative to baseline should not be interpreted as successful acquisition of the transformation. Instead, this pattern is more consistent with an instability or conflict regime during fine-tuning.
>
> At very low contamination, training is dominated by clean targets but includes a small fraction of systematically corrupted targets, which creates conflicting gradient signals over the same prompt distribution. The model can respond by making late-layer adjustments that partially accommodate the rare corrupted examples while still primarily optimizing for the clean objective. These adjustments can push the fine-tuned model’s late-layer geometry away from the baseline configuration, which CKA is sensitive to, even if the resulting changes are not yet coherent enough to produce the transformation reliably at generation time, hence low adherence. Notably, cosine similarity remains relatively high in this regime, consistent with disruption that affects cross-example geometry more than average token-level direction.
>
> As contamination increases, the supervision becomes more internally consistent. The model can then converge to a stable corrupted solution, and the late-layer representation space can become correspondingly more structured. In such a regime, CKA relative to baseline may evolve differently than in the low-contamination conflict regime, and need not decrease monotonically with contamination percentage. This provides a plausible explanation for why CKA can drop sharply at very low contamination even when behavioral adherence remains low, and then change non-monotonically as the model settles into a stable solution at higher contamination.
>
> **(5)** We agree that defense strategies are an important direction. The primary goal of this work is diagnostic, namely to systematically understand contamination vulnerabilities across model families, transformation types, and contamination levels. Developing and rigorously evaluating defenses is a substantial undertaking that warrants focused follow-up work. Importantly, while many of the components below are established practices, our contribution is to motivate which defenses are likely to be most relevant in this setting and where they should act, based on the syntactic versus semantic asymmetry, the low-contamination conflict regime, the capability curse, and the consistent localization of representational changes toward final blocks.
>
> - **Data validation and normalization for structural artifacts.** Since character-level corruption produces the most severe failures, dataset-side checks can serve as first-line defenses, such as Unicode normalization, character distribution heuristics, detection of systematic reversals, tokenization-level anomaly flags (e.g., unusually fragmented outputs), and filtering examples with unusually high edit distance or non-language character patterns.
>
> - **Contamination detection via lightweight filters.** Because our syntactic transformations are deterministic, reversal-like artifacts can be flagged with inexpensive rule-based checks. For semantic corruption, prompt–response mismatch can be screened using lightweight consistency signals (e.g., semantic alignment scoring or model-based screening for off-topic or mismatched responses), which can be used as pre-finetuning filters or as audits of instruction-tuning corpora.
>
> - **Robust training objectives and mixture control.** Our results show that even small fractions of systematic corruption can induce large behavioral shifts (including the low-contamination conflict regime), motivating (i) curriculum or reweighting strategies that down-weight suspicious samples, (ii) robust losses or gradient clipping to reduce sensitivity to rare high-gradient outliers, and (iii) early stopping or monitoring using held-out validation checks designed to detect contamination-like behavior.
>
> - **Output-side regularization anchored to baseline behavior.** Our representational findings indicate that contamination effects often localize toward the final blocks. This motivates targeted regularization during fine-tuning, such as penalties that constrain late-layer representations or the output distribution relative to the baseline model on clean validation prompts, with the goal of limiting contamination-induced drift while preserving earlier feature extraction.
>
> - **Scale-aware filtering for semantic contamination.** The capability curse result (Section 4.4) indicates that more capable models can acquire semantic corruptions with less exposure. This motivates stricter semantic consistency checks as model capability increases, and suggests that filtering thresholds calibrated on smaller models may be insufficient at larger scales.

---

> > ### Author Response · Authors · 2026-04-23
> > **Response to Review of Paper7981 by Reviewer fSmP (3/3)**
> >
> > **(6)** We provide architectural hypotheses for each family in Appendix G (Sections G.1–G.6). Briefly, Gemma3’s interleaving of local sliding-window and global attention layers may help explain the mid-stack CKA fluctuations; Phi4’s tied input/output embedding matrix is consistent with its distinctive early-layer sensitivity; OLMo2’s stability-oriented normalization and attention-logit control choices (e.g., post-norm placement and QK-norm) align with its near-ceiling CKA profiles; SmolLM2’s data-centric staged curriculum may contribute to its unusually stable intermediate-layer patterns; and Llama3.2’s homogeneous, uniform-attention decoder stack is consistent with smooth depth-wise cosine drift and minimal mid-network CKA structure. We emphasize that these are correlational hypotheses rather than causal conclusions, since each family differs in multiple factors (architecture, training data, and training recipe).
> >
> > **(7)** We agree that input side corruption and ‘input and output’ corruption are important extensions of our current design. Our main paper isolates output side contamination because it directly tests whether models internalize corrupted supervision as a generation behavior even when evaluation prompts are clean. To address this question directly, we ran additional ablations on Qwen2.5 models for syntactic transformations. We have presented the results of this ablation in Appendix H. We restrict this analysis to syntactic corruption because applying the semantic transformations on the input side would not introduce a qualitatively new corruption mode, as the semantic mismatch we already study on the output side would largely relocate to the prompt rather than define an orthogonal axis of variation.
> >
> > - **Input only corruption** differs qualitatively from output only corruption. When we corrupt only the input while keeping target outputs clean, the dominant effect is that the model is trained to map corrupted prompts back to correct answers. This tends to preserve grammatical correctness at near ceiling for character reversal inputs across all Qwen2.5 scales, while utility degrades mainly through loss of input intelligibility. Concretely, under character reversal input corruption, semantic similarity remains extremely low and accuracy remains near zero across contamination levels, while grammatical correctness stays around 99%. Under word reversal input corruption, utility remains substantially higher and increases with contamination level for all scales, with semantic similarity remaining high and accuracy rising with both scale and contamination level. This contrasts with our original output only setting, where character reversal in the output rapidly destroys well formedness and semantic similarity, indicating that output corruption directly contaminates the generation pathway, whereas input corruption primarily acts as a harder instruction following mapping problem.
> >
> > - When **both prompt and target** are corrupted within the same transformation, training no longer asks the model to recover clean outputs from corrupted inputs. Instead, it reinforces a corrupted mapping. In this regime, character reversal remains highly destructive in utility terms. Accuracy stays near zero across scales, semantic similarity remains low, and grammatical correctness collapses sharply because the model is trained to emit corrupted output. Sensitivity rises steeply as the contamination rate increases, indicating that the model internalizes the corruption pattern when it is present on both sides. For word reversal, ‘input and output’ corruption yields a different behavior from the output word reversal setting. Semantic similarity is substantially lower than the input only case, grammatical correctness is much lower than in the clean output setting, and sensitivity to contamination increases with contamination, indicating stronger internalization of the reversal pattern.
> >
> > - **Comparison to the original output only contamination.** Across the ablations, the key qualitative difference is where corruption is allowed to enter the generation pathway. Output corruption directly trains the model to emit corrupted strings, which is why character reversal causes severe breakdowns in grammatical correctness and semantic similarity. Input corruption instead trains the model to answer correctly despite corrupted inputs, so grammatical form can remain intact even when accuracy collapses. ‘Input and output’ corruption enforces a self consistent corrupted mapping and therefore tends to increase sensitivity to contamination and degrade output well formedness, especially for character reversal. Together, these results support the interpretation that output side supervision is the most direct driver of contamination induced behavioral failures, while input corruption leads to different failure modes that are better viewed as robustness to corrupted inputsrather than learning corrupted supervision.

---

### Author Response · Authors · 2026-04-23
**New Experiments and Paper Updates**

A common question across reviewers was how our results generalize when **contamination is introduced on the input side, or on both inputs and outputs**. We have updated the paper to address this and related comments, with **modifications highlighted in blue throughout the revised submission**.

- The main paper isolates output-side contamination to test whether corrupted supervision is internalized as a generation behavior even when evaluation prompts are clean. In response, we additionally ran syntactic ablations on the Qwen2.5 family under two extra settings that serve different diagnostic purposes. We have presented the results of this ablation in Appendix H.

- Input corruption tests robustness to corrupted prompts while keeping supervision correct, isolating whether the model can still follow the instruction when the input surface form is disrupted. 'Input and output' corruption tests whether the same transformation applied consistently to both sides changes the learning dynamics relative to our output-only setting, where the prompt is clean but supervision is corrupted.

- We restrict these ablations to syntactic transformations because applying the semantic transformations on the input side would not introduce a new corruption mode. For semantic corruption, the core issue is prompt-response mismatch. Moving that mismatch into the prompt largely relocates the same phenomenon rather than creating an orthogonal axis of variation.

- For each setting, we report the evaluation metrics used throughout the paper, namely accuracy, semantic similarity, and grammatical correctness. For the input-and-output settings, we additionally report sensitivity to contamination, since this metric measures adherence to the output-side transformation pattern, which is well-defined here but not in the input-only setting.

---

### Decision · Action_Editor_tGnF · 2026-05-31

**Recommendation:** Reject

**Audience:**

Yes

**Audience Explanation:**

The reviewing committee agrees that the paper addresses a practically important and timely problem with the deployment of SLMs increasing rapidly in resource constrained settings. The experimental scope is comprehensive covering multiple SLMs, contamination patterns, and contamination levels.

**Claims And Evidence:**

No

**Claims Explanation:**

- While the authors clearly justify their contribution as a diagnostic analysis of instruction tuning under controlled contamination patterns rather than a model of a specific adversary strategy, it will be important for readers to take away clear findings and practical guidance from an investigation on fine-tuning data contamination of SLMs for improving the applicability of real-world SLM deployments. The main motivation of the submitted work is the widespread deployment of SLMs in resource-constrained settings, hence it is important to keep the audience which will benefit most from this work to be at the forefront in terms of the clarity and applicability of the findings from such an investigation.
- The reviewing committee appreciates that authors release the GPT-4o generated test set for reproducibility. While the authors clarify that the GPT-4o generated test set is used primarily as a controlled, broad-coverage, and relatively easy evaluation distribution for the instruction-tuned SLMs in the submission, comparison with some commonly used benchmarks will still be helpful to ground the results in the context of prior work on SLMs by the community. It makes sense that the primary goal in this submission was to hold the evaluation constant while isolating how corrupted instruction-tuning supervision alters behavior relative to a baseline where models already perform strongly. However, even empirically testing and demonstrating the difficulty to evaluate on prior benchmarks will make the paper’s contributions and the justification for a new dataset stronger.
- The newly added input and input-output contamination comparisons are helpful but are only tested with Qwen 2.5 v/s the 23 SLMs tested for the output only contamination study in the paper. Testing the input related contaminations with a few more more SLM families (even if not the full set of 23 SLMs) will be helpful to the reader in terms of broader generalizability from the findings.
- The authors clarify in discussions with reviewers that the phenomena studied in their work are not claimed to be unique to SLMs; rather, their contribution is a systematic characterization within the SLM regime. However, a more thorough contextualization with similar prior work on LLMs or a comparison with open-source LLM families by the authors themselves will make the need for such a study on SLMs stronger.

**Resubmission Of Major Revision:**

The authors may consider submitting a major revision at a later time.